# Determination of Abrasiveness in Copper-Gold Sulfide Ores: A Contribution to the Geometallurgical Model of the Sossego Deposit

**Petterson de Azevedo Barbosa** [1,*] , **Maurício Guimarães Bergerman** [1,*] , **Elisabeth da Fonseca** [2] **and Rogerio Kwitko-Ribeiro** [2]

1   Department of Mining and Petroleum Engineering, Polytechnical School, University of São Paulo, São Paulo 05508-030, SP, Brazil

2   Mineral Development Center, Vale S.A., Santa Luzia 33040-900, MG, Brazil; elisabeth.fonseca@vale.com (E.d.F.); rogerio.kwitko@vale.com (R.K.-R.)

*   Correspondence: petterson.barbosa@usp.br (P.d.A.B.); mbergerman@usp.br (M.G.B.)

**Abstract:** The geological context of this study is established in the iron oxide-copper-gold (IOCG) deposit of Sossego (Canaã dos Carajás, Brazil), where hydrothermal alterations in shear zones concentrated the metals of interest and added new characteristics to the metavolcanic-sedimentary and granite rocks. The mineral transformation of rocks by hypersaline fluids enriched in metals and silica also modifies some metallurgical properties, such as abrasiveness. Special bench tests on rock drill cores are used in mapping the abrasiveness of rocks, with the Bond abrasion test being more commonly used in the mining industry, but it has a restrictive sampling protocol and mass requirement for geometallurgical studies. As a counterpoint, the test of the Laboratoire Central des Ponts et Chaussées/Central Laboratory of Bridges and Roads (LCPC) requires a smaller amount of fine material and a finer granulometric range. The study on the use of LCPC was implemented in 40 samples, using Bond Ai as a reference. The results showed a strong correlation between both methodologies ($R^2$ = 95%), validating the use of LCPC to quantify abrasiveness in the Sossego mine. It was also possible to classify the most abrasive lithologies.

**Keywords:** abrasiveness; geometallurgy; iron oxide-copper-gold ore; Bond abrasion test; LCPC

## 1. Introduction

Abrasion is the most common wear mechanism in ore mining and mineral processing operations [1–8]. Albertin and Sinatora [2] pointed out the relationship between the abrasiveness of minerals and the wear of metal surfaces when their investigations showed that high-chromium white cast iron mill balls underwent faster wear when processing quartz instead of iron ore and phosphate rocks. Piazzetta et al. [9] and Moradizadeh et al. [10] also found that materials with high equivalent quartz content caused higher abrasion wear on metal surfaces. Proper quantification of the abrasiveness of different ores may be based on standardized tests. Such quantification in the mining industry is usually made through the Bond [1] test method, the preferred alternative among the various abrasion tests available [2,4–7,11]. The outcome of this test is the Bond abrasion index (Bond Ai), which finds widespread use among metallurgists and suppliers of consumables for crushers and mills as a parameter to estimate the expected wear of such consumables [4,6].

The Bond Ai test requires 1.6 kg of material with particle size ranging from 19.05 to 12.7 mm (3/4 to 1/2 in). In certain cases, such as studies in the early stages of a project or geometallurgical studies, where only drill core fragments or drilling chips are available, it may be difficult to get such an amount of material within the required size range.

Among the various ore abrasion measuring methods described by Peres [11] (p. 12), the LCPC test stands out for requiring finer material (between 6.3 and 4.0 mm) and just 0.5 kg of the sample [12,13].

Specific protocols and machinery are used for such tests, although they preserve the same rock-metal interaction mechanism. Notably, the LCPC test uses a small sample volume, fine-grained in a more restrictive range, which is subjected to high rotation on a relatively soft metal plate. In other words, it is a test formatted to measure any minimal contact caused by minerals and is, therefore, very sensitive. Bond Ai test subjects a sample about three times larger in particle size, with a coarser and wider grain size, and to a smoother rotational speed to wear a harder plate than used in the LCPC. Thus, Bond's Ai wear plate has little effect in the case of soft rocks compared to the LCPC.

Despite the strong correlation between the abrasiveness results by Bond Ai and by LCPC described in the literature, its equation needs to be confirmed for different ores/mines, since the particularities of rock and mineral typologies, and even the difference between the steel plates used in different laboratories, can affect the correlation [11,14,15].

Geometallurgical studies are widely used to help operations improve their profits [16–19]. For the Sossego mine, which includes the open pits—Sossego, Sequeirinho, and Pista—and an ore processing plant, geometallurgical studies [20–23] were carried out, mainly during the pre-feasibility and feasibility stages, extending over the first years of production (2004), until 2017. This historical geometallurgical database, totaling 135 samples, was obtained from exploratory drilling holes and submitted to bench tests for technological characterization of typical ores to support the definition of the appropriate process route, as well as the dimensioning of mine equipment for operational studies. Figure 1a illustrates the location of the total samples by lithology and by mining plan year for these geometallurgical historical data. The gray object in image "b" is the reference topography of December 2017.

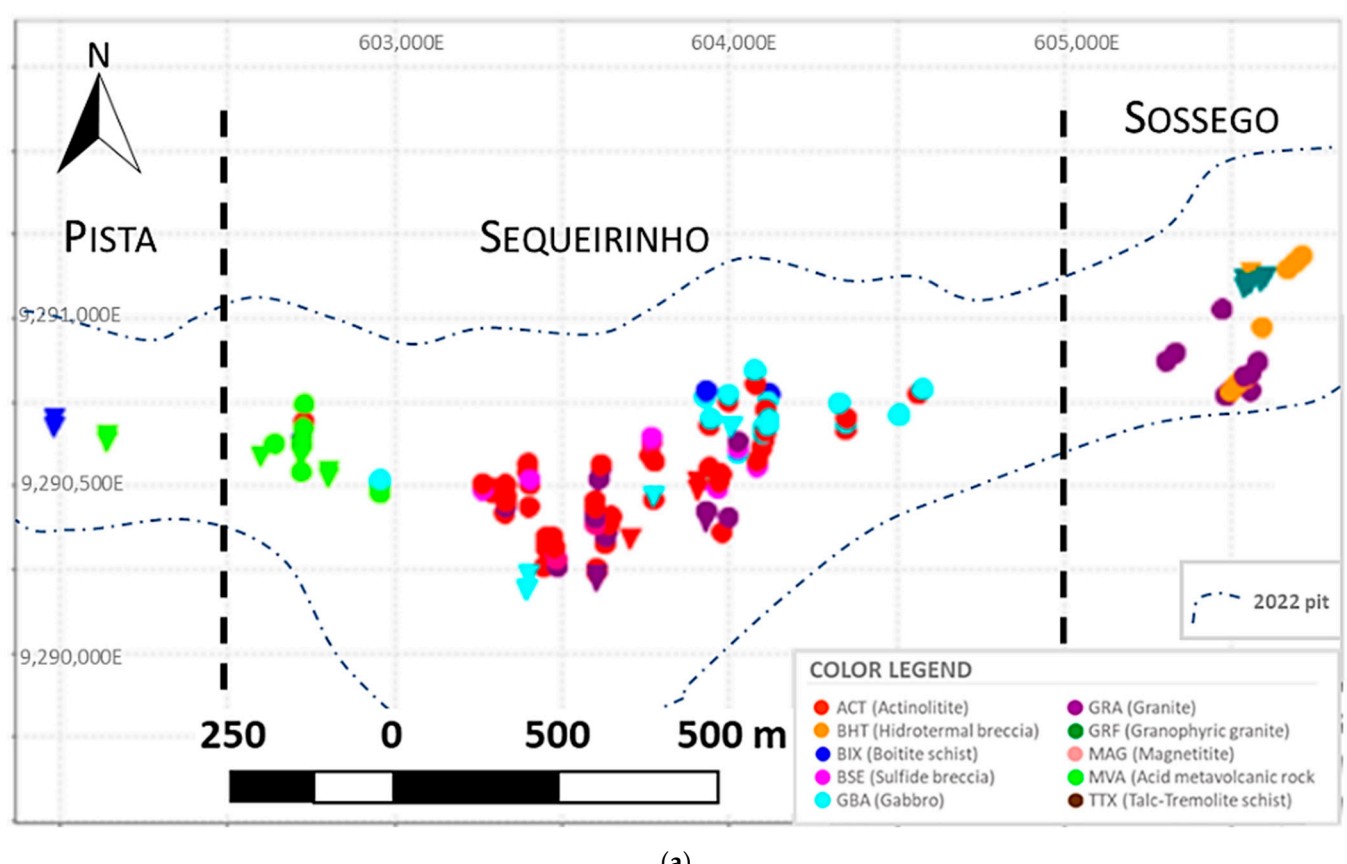

(a)

**Figure 1.** *Cont.*

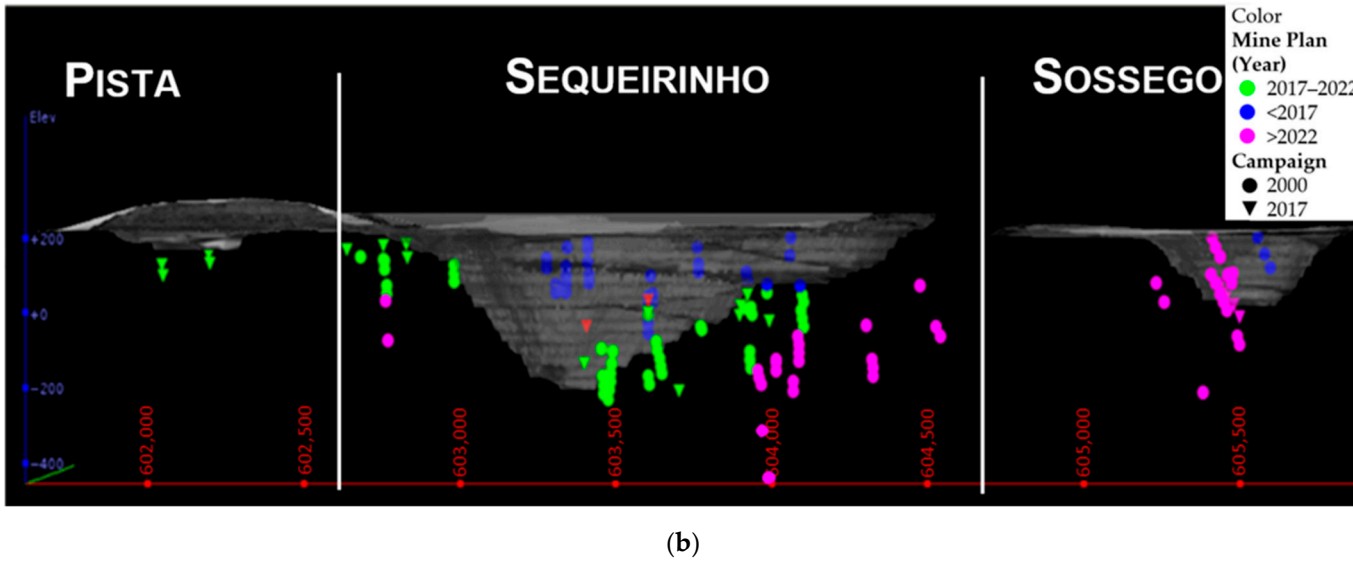

(**b**)

**Figure 1.** (**a**) 2D location map for Sossego's historical data with 135 samples at the Pista, Sequeirinho, and Sossego pits by lithology; (**b**) 3D view E–W to North direction of samples location by year of mining plan. The gray topography represents the pit as in 2017. The circle and triangle symbols on sample positions represent the two old campaigns carried out in 2000 and 2017, respectively.

These past studies at Sossego Mine [22] include a qualitative and quantitative mineralogy description, drop weight test (DWT), point load test (PLT), tumbler index (ta), SATMAGAN® (magnetic susceptibility), Bond work index (Wi), and the Bond abrasiveness index (Ai). The following lithology codes were used based on Vale procedures: ACT (actinolitite), BHT (hydrothermal breccia), BIX (biotite schist), BSE (Sequeirinho breccia), BSO (Sossego breccia), GBA (gabbro), GRA (granite), GRF (granophyric granite), MAG (magnetitite), MVA (metavolcanic acid), and TTX (talc-tremolite schist).

Only 60 samples were tested for Bond Ai, and a large range of abrasiveness values were found (Figure 2). A low representativity across the pits was considered due to not having samples on the Pista area between 2022–2024 (final pit) and having only a few samples at the bottom of Sequeirinho, until its lifetime in 2026. The low number of samples and the large range in values provides an opportunity to investigate this property again with new samples and methods to formulate new understandings. For the historical data [22], the MAG and TTX lithology groups were not tested by the Bond Ai and the breccia samples were distinguished by copper-gold sulfide ore from Sossego body (BSO), Sequeirinho body (BSE), and without ore (BHT).

In general, the graph in Figure 2 represents what is expected of the abrasive behavior of the rocks of the Sossego deposit, with a high variation of this property among the studied rocks. Although the MVA lithology had the highest abrasiveness average, some other lithologies had no expected high values, such as granitic types (GRA and GRF). The high silica [22–24] content is an answer for the MVA group, and almost all its occurrences are localized in a portion of Pista pit, at the west trend of the Sequeirinho body. Similar to the high values, the large ranges observed at almost all lithological groups were unsatisfactory for the establishment of future effective use. It caused a strong negative impact between January 2017 to the end of 2019, specifically for MVA processing. These feed changes resulted in a high cost in wear material consumption, increasing by 7% over the wear of SAG and ball mills screens and crusher liners. All these aspects made it necessary to carry out a new abrasiveness study for the Sossego deposit.

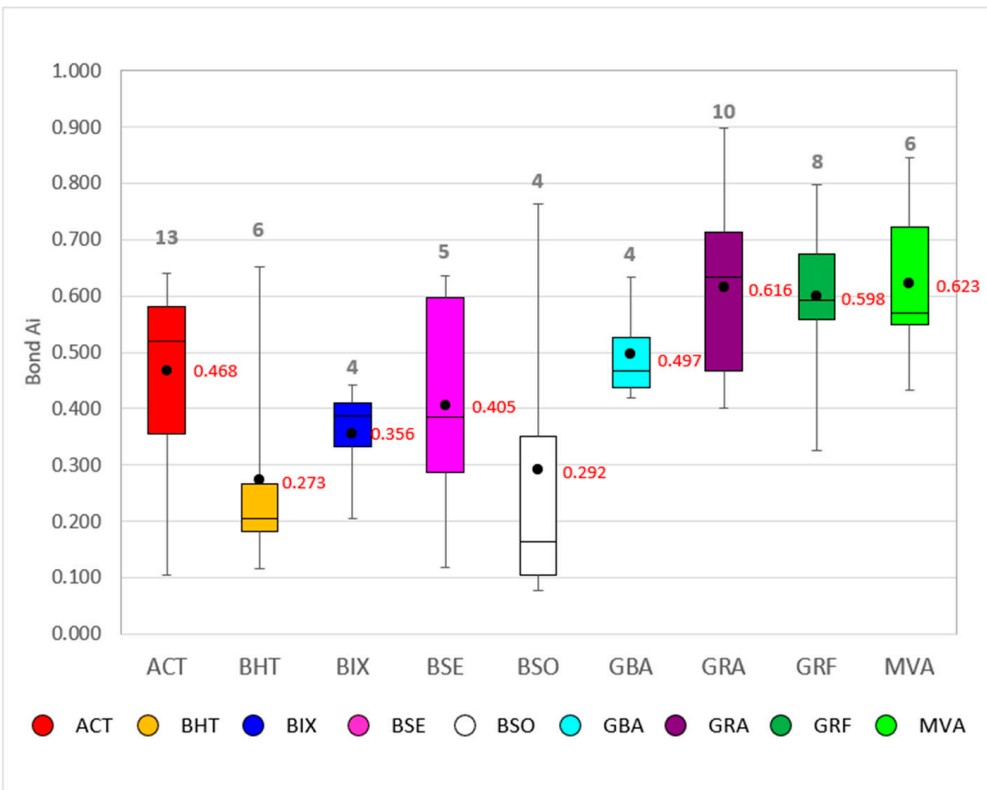

**Figure 2.** Statistic box plot graph of Bond Ai values with the sample quantity (except for MAG and TTX groups). The colors are standardized by lithology. The values above each bar are the number of samples and in the right position are the average.

In this study, 40 new samples from different areas of the Sossego, Sequeirinho, and Pista pits were characterized using the Bond and LCPC methodologies. The main objective was to validate the correlation between them so that the simplified LCPC test could be used for future geometallurgical studies and to identify the lithologies and mine domains with higher abrasion values.

No comparative study, including the historical data, could be carried out regardless of the authors' choice, given the differences in the type of samples used and uncertainties about the lithological classification protocols and the sample handling preparation applied for the Bond Ai of historical geometallurgical campaigns.

## 2. Materials and Methods

According to the planning of this sampling campaign, 40 new samples were randomly collected, representing the lithological diversity of the mineral deposit and the study's objectives. These samples were identified with the initials AMPT, followed by a sequential number. Thus, the eight main lithologies distributed in three pits were considered, as seen in the map in Figure 3. Some of these samples were collected from blasted outcrops inside the pits, represented on the map by yellow points, and another group of samples was collected from temporary ore stockpiles at red points.

The lithologies were coded, following the same procedures described in Section 1: ACT, BIX, BSE, GRA, GRF, MAG, MVA, and TTX. The MVA was preferentially sampled based on the preliminary information from the processing plant about its high abrasiveness. However, the campaign comprised only two samples for ACT because its occurrence was limited. The BHT of historical data was the same texture and mineral assemblage as BSE in this study.

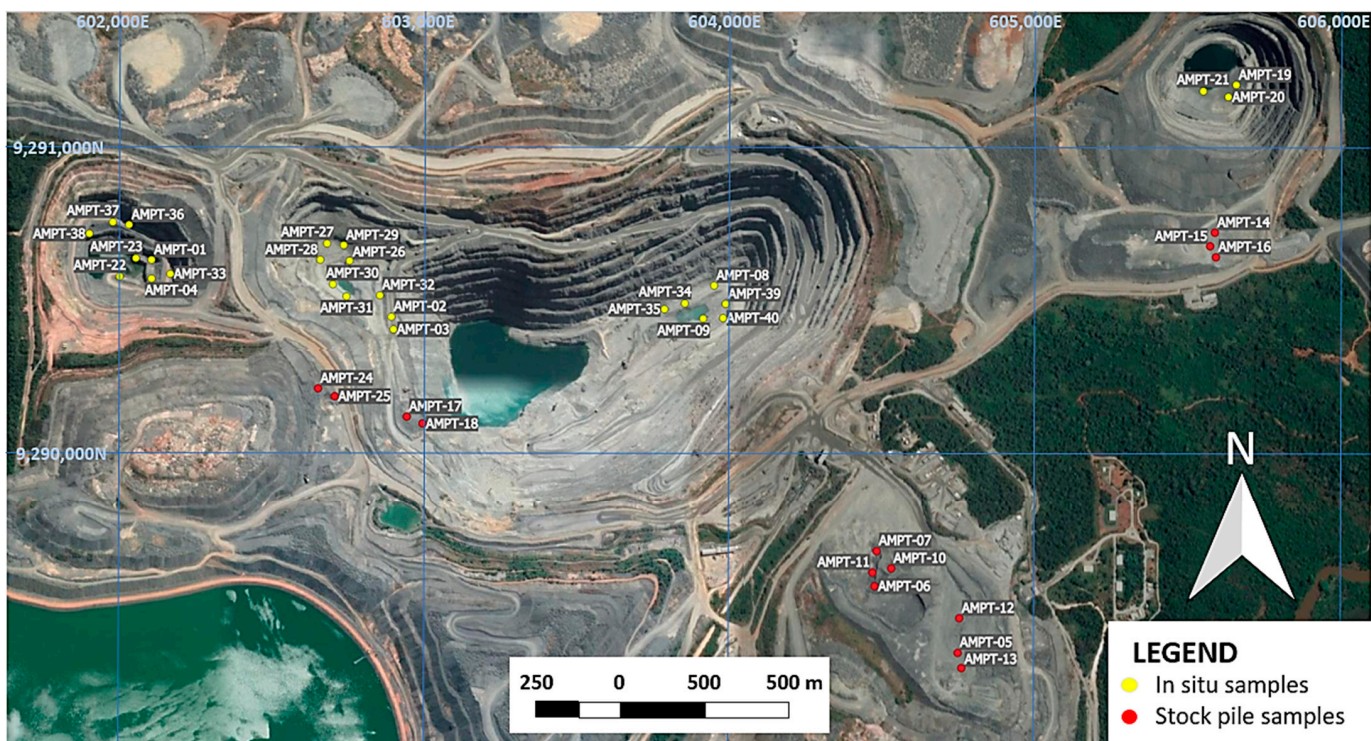

**Figure 3.** Location map of 40 new samples. Yellow targets for in situ samples and the red targets for samples transported from their origin [25]. Reference image from 07 August 2021.

The samples, weighing around 5–7 kg and between 75 and 10 mm in size, were stage crushed in a jaw crusher until 100% below 19.05 mm and classified between 19.05 and 12.7 mm. Four 400 g aliquots were prepared from each sample for the Bond test and one 1 kg aliquot was taken for the LCPC test, ensuring that both tests would be conducted on mineralogically similar materials. The LCPC aliquots were subsequently stage crushed in a jaw crusher until 100% below 6.3 mm and above 4.0 mm.

The LCPC test was carried out according to the French Standard P18–579 [13]. The test measures the mass loss of a metal plate with 50 mm × 25 mm × 5 mm in size that rotates for 5 min at 4500 rpm in contact with 500 g of material at the specified size range between 6.3 and 4.0 mm. The plate is made from low carbon steel (C1015), with Rockwell hardness from 60 HRB to 75 HRB. The LCPC abrasiveness index is determined according to Equation (1):

$$LCPC\ (g/t) = 1000 \times 1000\ (Mip - Mfp)/M, \tag{1}$$

where:

Mip—plate mass before to the LCPC test;
Mfp—plate mass after the LCPC test;
M—sample mass (500 ± 0.2 g)

The Bond test follows the protocol proposed by Bond [1], according to which four 400 g samples are processed for 15 min in a Bond's abrasion tester. The material abrasiveness is calculated from the wear of a 500-Brinel hardened SAE 4325 steel plate/implement fitted inside the equipment according to Equation (2).

$$Ai\ (g) = Mi - Mf, \tag{2}$$

where:

Ai—Bond abrasion index;
Mi—initial steel plate mass;
Mf—final steel plate mass.

The hardness of all metal plates was measured using a Rockwell Hardness Tester (manufactured by Wilson, New York, NY, USA) to ensure their hardness met the applicable requirements [1,13]. Such measurement indicated that the LCPC test plates had a 65 HRB average hardness, falling within the range as defined by the French standard [13]. The hardness test result for the plates used in the Bond test was 525 HB, close to the 500 HB hardness specified by Bond [1]. The plates were from the same lot and manufacturer in all tests, aimed at preventing possible manufacturing variations from affecting the test results. The mineralogy of the final product ground was measured to confirm the similarity between the samples used in LCPC and Bond Ai and used to classify lithologies, sample by sample.

For the mineralogical characterization, polished sections were produced, following a proprietary method of cold epoxy embedding under centrifugation [26]. The preparation with sample to epoxy ratio of 1:2 (vol.) was centrifuged, demolded, cut along the vertical axis, and potted again in a 30 mm round mold to ensure the representativeness of a single surface in terms of morphology, size, density, and particles' composition. The modal mineralogy was generated by the system QEMSCAN® (Quanta 650W, manufactured by FEI, Brisbane, Australia) of SEM-based automatized mineralogy, consisting of a FEI Quanta 650 SEM (manufactured by FEI, Brisbane, Australia) with two Bruker XFlash 6|30 EDS (manufactured by Bruker, Brisbane, Australia) detectors. All measurements were produced in the Field Image mode, under 25 kV of beam acceleration and 10 nA of sample current, using 15 μm × 15 μm of pixel spacing and 1500 X-ray counts per pixel. An average of 2 million pixels per sample were then generated, 30% of which were identified as mineral phases.

The chemistry of major elements was determined by infrared absorption, thermal conductivity with direct combustion (S), and inductively coupled plasma (ICP-OES, model, manufactured by Agilent, model5110, Victoria, Australia) with solubilization in aqua regia (Cu), solubilization in $HNO_3$ + HF (K and Na), and calcination at 600 °C with fusion in $Na_2CO_3/Na_2B_4O_7$ (Ca, Fe, and Si).

## 3. Results

The results comprise an initial and determinant validation that aims to guarantee the tests reproducibility and the similarity between the pairs of the 40 new tested samples. The statistical study defined the lithological composition of the samples and mathematically represented the abrasiveness property.

### 3.1. Validations

A quality control campaign was implemented using five duplicate samples to evaluate the reproductivity of the Bond Ai and LCPC tests. Only the sample AMPT-30 showed a deviation greater than 15%; however, this was not significant due to their low values at around zero. All the samples were within the expected range, below 10% and, therefore, acceptable (Table 1).

The preliminary visual lithological classification was confirmed through mineralogical evaluation. Assay reconciliation (Figure 4) allows to compare QEMSCAN® calculated chemical data obtained from mineral chemistry with externally measured chemical data and is used for validation of mineralogical data. The data are plotted against one another on the chart for visual comparison and identification of potentially anomalous measurements. A slight dispersion from 1:1 regression, as observed in the graph, is expected and is due to fluctuations in mineral chemistry from an average stoichiometry.

The small cloud dispersion and the trendline inclination (Figure 5), using the LCPC and Bond Ai values, could reinforce the effectiveness of the sample preparation methodology using just the six major mineral types. These analyses satisfied the primary requirement of similarity between the tested pairs and confirmed the lithological classifications.

**Table 1.** Results of duplicate samples.

| Sample | Test Type | Samples No. | Average | Relative Error |
|--------|-----------|-------------|---------|----------------|
| AMPT-11 | Bond Ai | 3 | 0.490 | 8.16% |
| | LCPC | 3 | 1192 | 2.35% |
| AMPT-18 | Bond Ai | 2 | 0.245 | 6.12% |
| | LCPC | 2 | 830 | 1.75% |
| AMPT-21 | Bond Ai | 2 | 0.225 | 2.22% |
| | LCPC | 2 | 863 | 5.91% |
| AMPT-23 | Bond Ai | 2 | 0.700 | 0.00% |
| | LCPC | 2 | 1441 | 0.94% |
| AMPT-30 | Bond Ai | 2 | 0.015 | 33.33% |
| | LCPC | 2 | 111 | 10.41% |

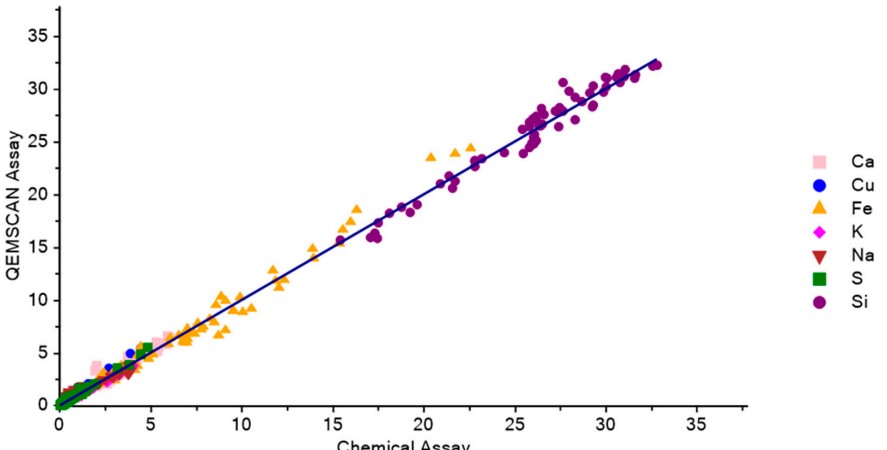

**Figure 4.** QEMSCAN® assay reconciliation of the 72 measured samples. The 1:1 trendline is in blue.

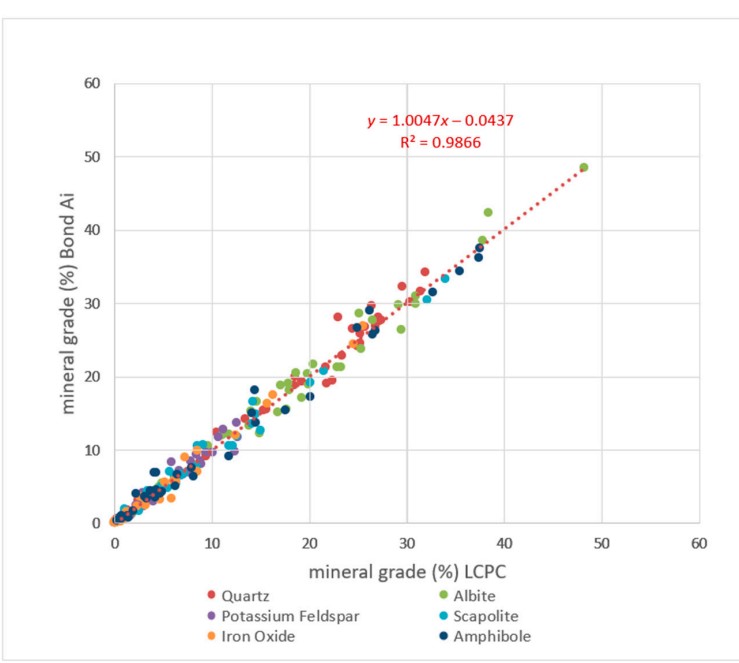

**Figure 5.** Mineralogy results for the six major mineral groups sampled for LCPC and Bond Ai.

Charts for each pair of samples in Supplementary Materials (Figures S1–S36) summarize graphs of mineralogical analysis from QEMSCAN$^{®}$ data used for lithological classification, such as horizontal bars and scatterplot of the mineral percent, in addition to microscopic images with the mineralogical assemblage. The first four samples were not part of the analysis, maintaining the mineralogical field classification.

### 3.2. Lithological Study

The database in Table 2 includes the 40 new samples, each one with its coded lithology, distinguished by two sets of raw values of abrasiveness (non-normalized measured values) acquired in these tests. The results show a wide variation of abrasiveness, compatible with the expected hardness for the analyzed lithologies, whereas the value ranges are from 0.006 to 0.770 for Bond Ai and from almost 0 to 1490 for LCPC, suggesting high abrasiveness.

**Table 2.** List of sample results and lithological classification.

| Sample | Lithology | Bond Ai | LCPC | Sample | Lithology | Bond Ai | LCPC |
|---|---|---|---|---|---|---|---|
| AMPT-01 | MVA | 0.674 | 1394 | AMPT-21 | GRF | 0.220 | 914 |
| AMPT-02 | MVA | 0.532 | 1278 | AMPT-22 | MVA | 0.770 | 1490 |
| AMPT-03 | MVA | 0.545 | 1269 | AMPT-23 | MVA | 0.700 | 1427 |
| AMPT-04 | MVA | 0.520 | 1304 | AMPT-24 | BIX | 0.260 | 768 |
| AMPT-05 | MVA | 0.260 | 939 | AMPT-25 | MVA | 0.510 | 1203 |
| AMPT-06 | MVA | 0.510 | 1143 | AMPT-26 | MVA | 0.570 | 1287 |
| AMPT-07 | ACT | 0.390 | 956 | AMPT-27 | BIX | 0.280 | 849 |
| AMPT-08 | ACT | 0.330 | 817 | AMPT-28 | MVA | 0.580 | 1289 |
| AMPT-09 | BSE | 0.340 | 941 | AMPT-29 | MAG | 0.370 | 850 |
| AMPT-10 | MVA | 0.420 | 1035 | AMPT-30 | TTX | 0.020 | 99 |
| AMPT-11 | MVA | 0.490 | 1222 | AMPT-31 | TTX | 0.006 | 31 |
| AMPT-12 | BIX | 0.210 | 731 | AMPT-32 | MVA | 0.720 | 1379 |
| AMPT-13 | BSE | 0.370 | 1089 | AMPT-33 | MVA | 0.620 | 1395 |
| AMPT-14 | GRA | 0.310 | 981 | AMPT-34 | BIX | 0.280 | 808 |
| AMPT-15 | GRA | 0.330 | 966 | AMPT-35 | BIX | 0.160 | 656 |
| AMPT-16 | GRA | 0.380 | 970 | AMPT-36 | BIX | 0.100 | 405 |
| AMPT-17 | MVA | 0.600 | 1303 | AMPT-37 | BIX | 0.140 | 685 |
| AMPT-18 | BSE | 0.230 | 844 | AMPT-38 | MAG | 0.360 | 874 |
| AMPT-19 | GRF | 0.250 | 813 | AMPT-39 | MAG | 0.430 | 1037 |
| AMPT-20 | GRF | 0.270 | 884 | AMPT-40 | BIX | 0.240 | 914 |

A visualization of all datasets is presented in the box plot of Figure 6 below based on the lithology variable. Most of the lithologies do not have extensive tails, and the most discrepant outliers are in the MVA lithology, which is explained by its heterogeneity and mineral assemblage. The layers formed on the MVA, with felsic bands of dominant quartz and feldspar at odds with mafic bands composed mainly of amphibole/actinolite, biotite, and scapolite, could represent this variability.

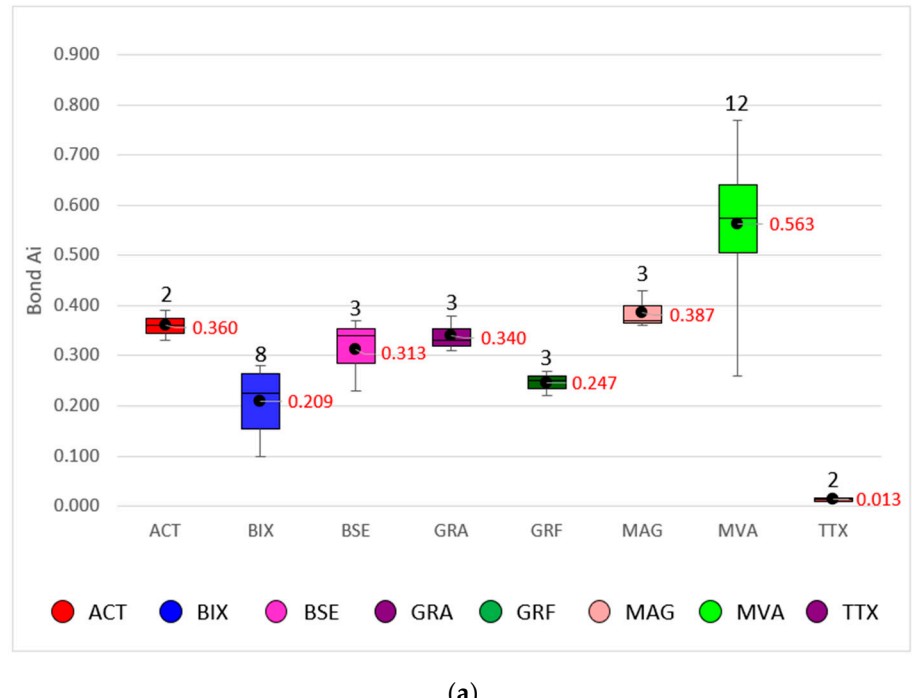

(**a**)

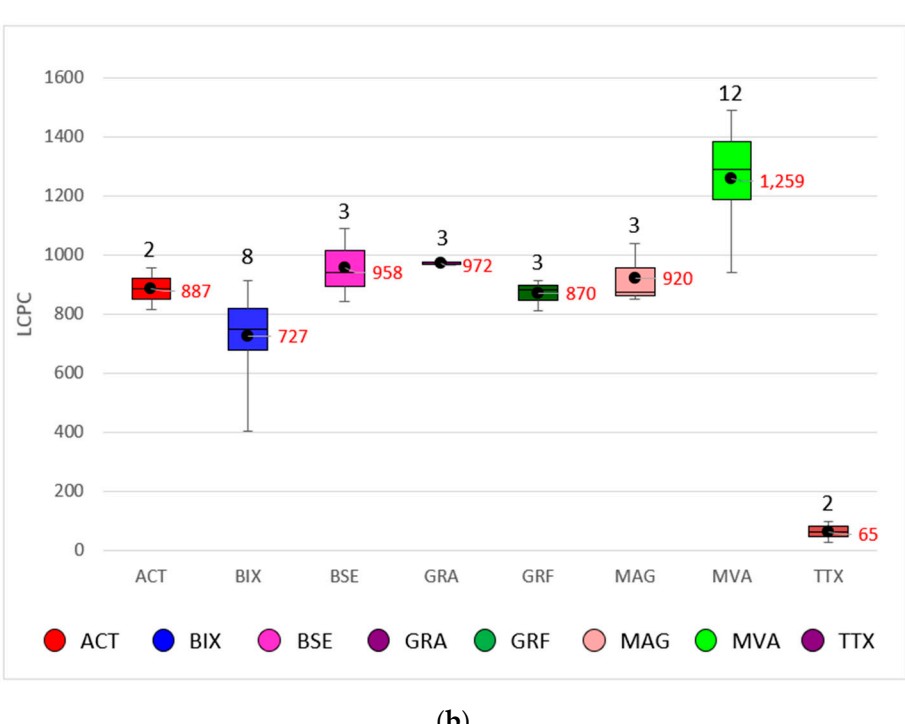

(**b**)

**Figure 6.** Results from new abrasiveness datasets with box plot by lithology for (**a**) Bond Ai; (**b**) LCPC. The colors are standardized by lithology. The values above each bar are the number of samples and in the right position are the average.

### 3.3. Statistical Analysis

The statistical summary by lithology is shown in Table 3. The TTX (high talc) lithology had the lowest values for Bond Ai (mean and median of 0.013) and LCPC (mean and median of 65), with the highest values seen in the MVA lithology (mean and median of Bond Ai around 0.570 and LCPC around 1288).

**Table 3.** Statistical summary of test results for the entire sample population and by lithology.

| | Stats | Bond Ai | LCPC | | Stats | Bond Ai | LCPC | | Stats | Bond Ai | LCPC |
|---|---|---|---|---|---|---|---|---|---|---|---|
| | Global | 40 | 40 | BSE | Count Num | 3 | 2 | MAG | Count Num | 3 | 2 |
| | Minimum | 0.006 | 31 | BSE | Minimum | 0.230 | 844 | MAG | Minimum | 0.360 | 850 |
| | Maximum | 0.770 | 1490 | BSE | Maximum | 0.370 | 1089 | MAG | Maximum | 0.430 | 1037 |
| | Mean | 0.382 | 981 | BSE | Mean | 0.313 | 958 | MAG | Mean | 0.387 | 920 |
| | Median | 0.365 | 961 | BSE | Median | 0.340 | 941 | MAG | Median | 0.370 | 874 |
| | Stand Dev | 0.189 | 327 | BSE | Stand Dev | 0.074 | 123 | MAG | Stand Dev | 0.038 | 102 |
| | Coeff var | 49.52 | 33.30 | BSE | Coeff var | 23.52 | 12.88 | MAG | Coeff var | 9.79 | 11 |
| ACT | Count Num | 2 | 2 | GRA | Count Num | 4 | 2 | MVA | Count Num | 15 | 2 |
| ACT | Minimum | 0.330 | 817 | GRA | Minimum | 0.310 | 966 | MVA | Minimum | 0.260 | 939 |
| ACT | Maximum | 0.390 | 956 | GRA | Maximum | 0.420 | 1035 | MVA | Maximum | 0.770 | 1490 |
| ACT | Mean | 0.360 | 887 | GRA | Mean | 0.360 | 988 | MVA | Mean | 0.573 | 1288 |
| ACT | Median | 0.360 | 887 | GRA | Median | 0.355 | 976 | MVA | Median | 0.570 | 1289 |
| ACT | Stand Dev | 0.042 | 98 | GRA | Stand Dev | 0.050 | 32 | MVA | Stand Dev | 0.122 | 133 |
| ACT | Coeff var | 11.78 | 11.09 | GRA | Coeff var | 13.80 | 3.245 | MVA | Coeff var | 21.25 | 10.30 |
| BIX | Count Num | 8 | 8 | GRF | Count Num | 3 | 3 | TTX | Count Num | 2 | 2 |
| BIX | Minimum | 0.100 | 405 | GRF | Minimum | 0.220 | 813 | TTX | Minimum | 0.006 | 31 |
| BIX | Maximum | 0.280 | 914 | GRF | Maximum | 0.270 | 914 | TTX | Maximum | 0.020 | 99 |
| BIX | Mean | 0.208 | 727 | GRF | Mean | 0.247 | 870 | TTX | Mean | 0.013 | 65 |
| BIX | Median | 0.225 | 750 | GRF | Median | 0.250 | 884 | TTX | Median | 0.013 | 65 |
| BIX | Stand Dev | 0.068 | 155 | GRF | Stand Dev | 0.025 | 52 | TTX | Stand Dev | 0.010 | 48 |
| BIX | Coeff var | 32.73 | 21.35 | GRF | Coeff var | 10.20 | 5.96 | TTX | Coeff var | 76.15 | 73.97 |

A principal component analysis was on Figure 7 and performed to evaluate the mineralogical assemblage and abrasiveness behavior (LCPC), where it can be seen that, in addition to quartz, other minerals showed a strong correlation with abrasiveness, such as albite, biotite, talc, and scapolite.

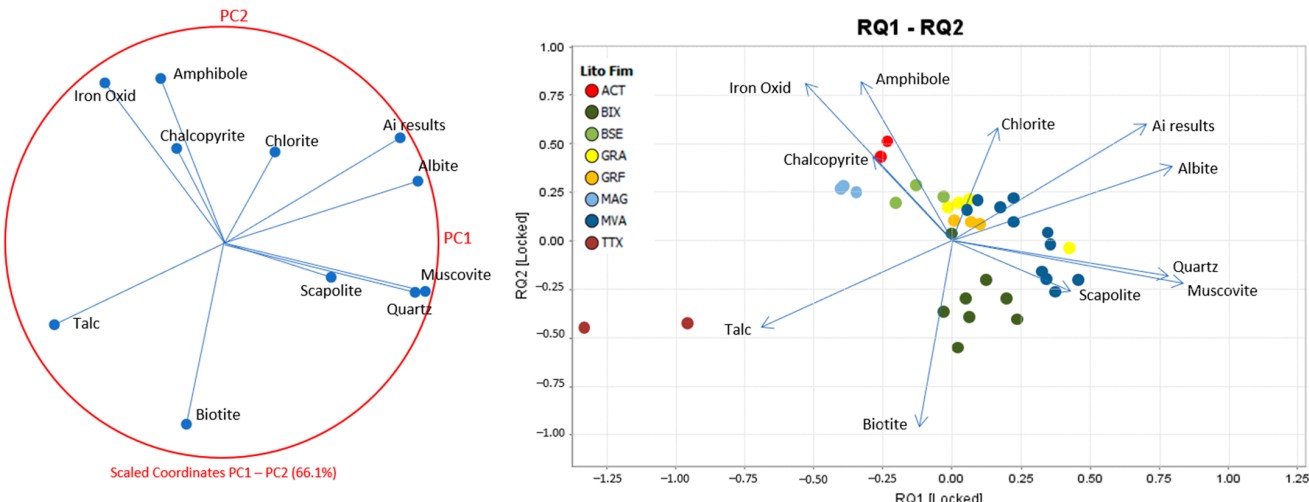

**Figure 7.** Principal component analysis of minerals categorized by lithology from ioGAS™ software.

Table 4 shows a correlation matrix between the 12 minerals and the LCPC values, in which the highest correlation could be seen at a value of 0.73 for albite, followed by scapolite, muscovite and quartz. It still confirms the low correlation with the talc mineral.

**Table 4.** Correlation matrix from ioGAS™ software [27] showing the correlation between the minerals and LCPC.

| Correlation | LCPC | Quartz | Albite | K-feldspar | Muscovite | Scapolite | Biotite | Chlorite | Talc | Amphibole | Fe-Oxides | Apatite | Carbonate |
|---|---|---|---|---|---|---|---|---|---|---|---|---|---|
| LCPC | 1 | 0.37 | 0.73 | 0.29 | 0.38 | 0.52 | −0.64 | 0.033 | −0.68 | 0.033 | −0.087 | −0.071 | 0.074 |
| Quartz | 0.37 | 1 | 0.35 | 0.58 | 0.6 | 0.12 | 0.08 | 0.39 | −0.42 | −0.75 | −0.61 | −0.55 | 0.43 |
| Albite | 0.73 | 0.35 | 1 | 0.33 | 0.43 | 0.1 | −0.45 | 0.1 | −0.42 | −0.22 | −0.38 | −0.24 | 0.037 |
| K-feldspar | 0.29 | 0.58 | 0.33 | 1 | 0.42 | −0.2 | −0.3 | 0.58 | −0.37 | −0.35 | −0.33 | −0.062 | 0.53 |
| Muscovite | 0.38 | 0.6 | 0.43 | 0.42 | 1 | 0.44 | 0.17 | −0.16 | −0.36 | −0.54 | −0.62 | −0.45 | −0.14 |
| Scapolite | 0.52 | 0.12 | 0.1 | −0.2 | 0.44 | 1 | −0.12 | −0.57 | −0.21 | −0.064 | −0.049 | −0.19 | −0.37 |
| Biotite | −0.64 | 0.08 | −0.45 | −0.3 | 0.17 | −0.12 | 1 | −0.29 | 0.24 | −0.35 | −0.35 | −0.31 | −0.24 |
| Chlorite | 0.033 | 0.39 | 0.1 | 0.58 | −0.16 | −0.57 | −0.29 | 1 | −0.24 | −0.14 | −0.14 | 0.13 | 0.7 |
| Talc | −0.68 | −0.42 | −0.42 | −0.37 | −0.36 | −0.21 | 0.24 | −0.24 | 1 | −0.17 | 0.046 | −0.16 | −0.14 |
| Amphibole | 0.033 | −0.75 | −0.22 | −0.35 | −0.54 | −0.064 | −0.35 | −0.14 | −0.17 | 1 | 0.73 | 0.68 | −0.21 |
| Fe-Oxides | −0.087 | −0.61 | −0.38 | −0.33 | −0.62 | −0.049 | −0.35 | −0.14 | 0.046 | 0.73 | 1 | 0.48 | −0.12 |
| Apatite | −0.071 | −0.55 | −0.24 | −0.062 | −0.45 | −0.19 | −0.31 | 0.13 | −0.16 | 0.68 | 0.48 | 1 | −0.083 |
| Carbonate | 0.074 | 0.43 | 0.037 | 0.53 | −0.14 | −0.37 | −0.24 | 0.7 | −0.14 | −0.21 | −0.12 | −0.083 | 1 |

### 3.4. Mathematical Correlation

The relationship between Bond Ai and LCPC can be studied through dispersion diagrams, which are used to demonstrate this correlation. Two trendlines in Figure 8 represent different correlations. The data lie closer to a quadratic polynomial equation (blue) than a linear trend (red), increasing the determination coefficient ($R^2$) from 87.70% to 94.57%.

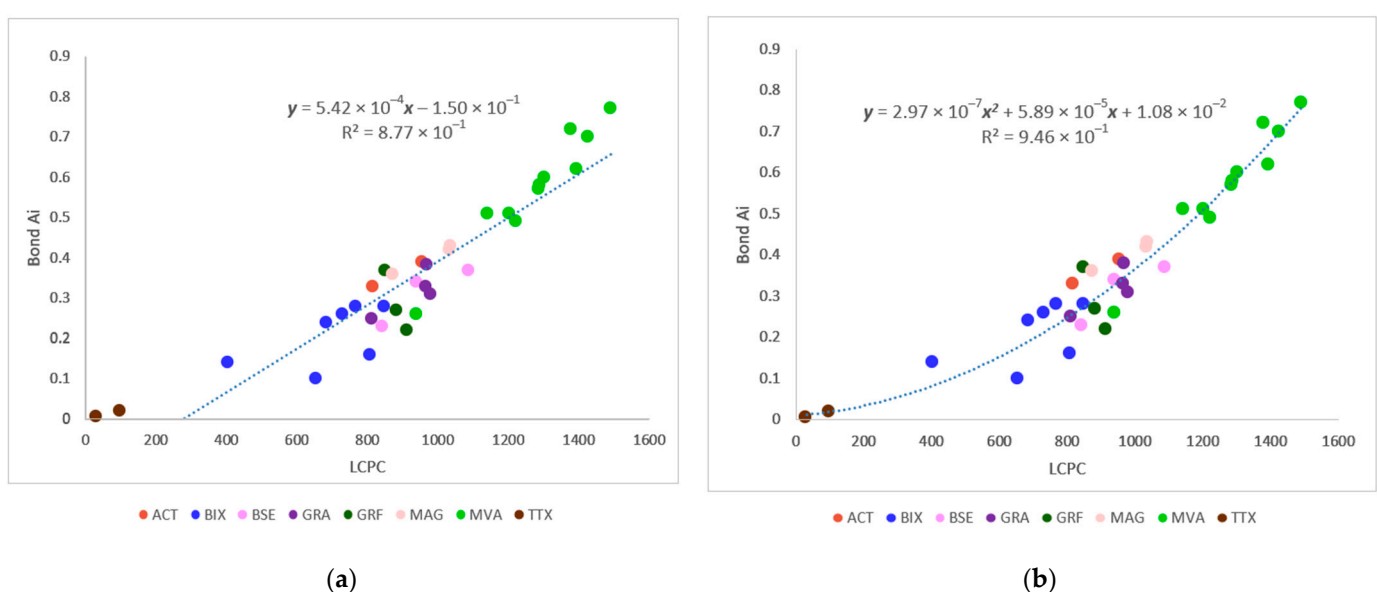

**Figure 8.** Correlation graphs LCPC versus Bond Ai for the total set of samples from the Sossego deposit. The dots are colored according to their lithologies and expose different correlations: (**a**) linear and (**b**) quadratic.

Table 5 for a summary of statistical results including analysis of variance to the regression and to the predictive model for Bond Ai based on LCPC values shows a constant variance for residues, and the zero *p*-value, validating the equation.

**Table 5.** Statistic output table with summary of equation results and the residue analysis for the predictive model for Bond Ai based on LCPC values from Minitab® software [28].

| Model Summary | | | Analysis of Variance | | | | | | Analysis of Variance | | | | |
|---|---|---|---|---|---|---|---|---|---|---|---|---|---|
| **S** | **R-sq** | **R-sq(ad)** | **Source** | **DF** | **SS** | **MS** | **f-Value** | **p-Value** | **Source** | **DF** | **SS** | **f-Value** | **p-Value** |
| 0.0449929 | 94.57% | 94.24% | Regression | 4 | 1.16399 | 0.581995 | 287.50 | 0.000 | Linear | 1 | 1.07937 | 242.36 | 0.000 |
| – | – | – | Error | 33 | 0.06680 | 0.002024 | – | – | Quadratic | 1 | 0.08462 | 41.80 | 0.000 |
| – | – | – | Total | 35 | 1.23080 | – | – | – | – | – | – | – | – |

A residue analysis was performed to prove the quadratic correlation, observing a normal distribution and a symmetric histogram (Figure 9).

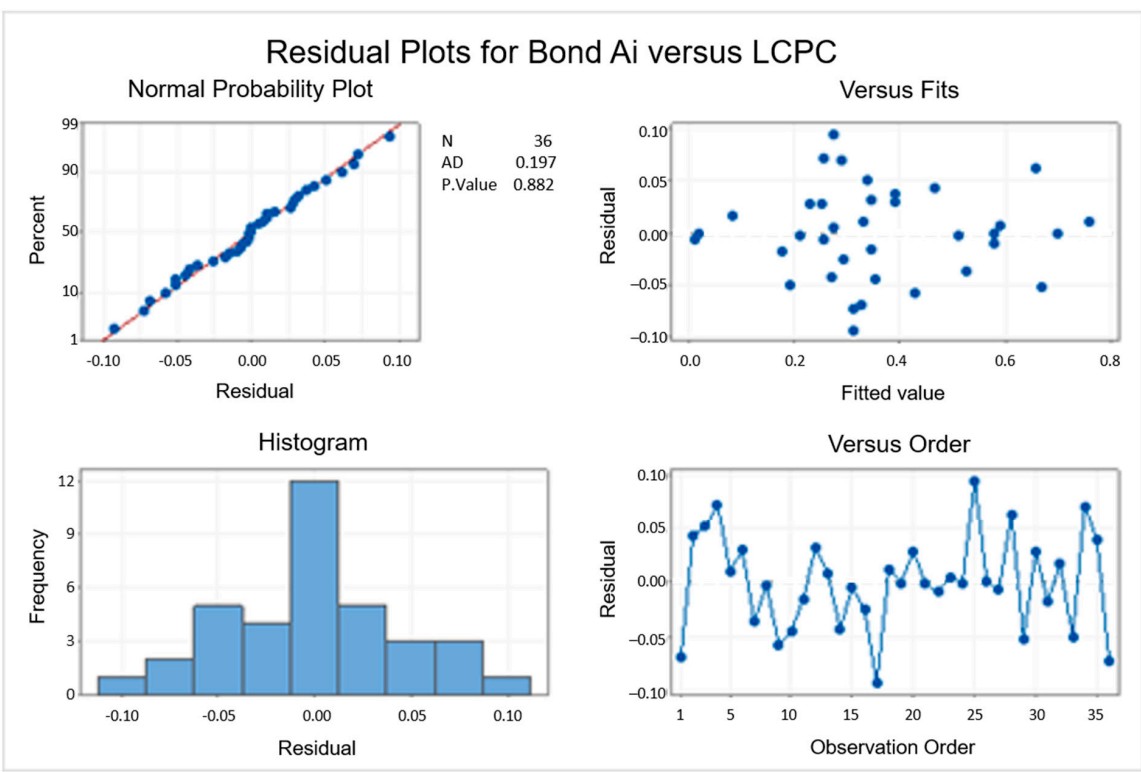

**Figure 9.** Residue graphs from Bond Ai versus LCPC at normal fitted and a symmetric histogram from Minitab® software [28].

According to the validations for this database, both tests (Bond Ai and LCPC) represent the same phenomenon of abrasion wear with a high (above 94.57%) correlation index.

Therefore, the Bond Ai forecast for a given value of LCPC for the copper-gold of the Sossego deposit can be described by the following Equation (3):

$$Y = 2.97 \times 10^{-7}x^2 + 5.89 \times 10^{-5}x + 1.08 \times 10^{-2}, \mathrm{R}^2 = 0.95, \tag{3}$$

where:

$Y$—Bond Ai value;

$x$—LCPC value;

An additional regression analysis was done to predict the LCPC value from the mineral content, establishing their direct correlation. A multiple regression Equation (4) was obtained to estimate consideration of the most significant minerals, such as quartz, biotite, albite, and chlorite. They were transformed into logarithms, as shown below (Table 6).

$$\mathrm{LCPC} = 980 - 446 \log.\mathrm{biotite} + 286 \log.\mathrm{albite} - 427 \log.\mathrm{chlorite} + 333 \log.\mathrm{quartz}, \tag{4}$$

where:

biotite—biotite mineral grade;
albite—albite mineral grade;
chlorite—chlorite mineral grade;
quartz—quartz mineral grade.

**Table 6.** Multiple regression to predict the LCPC values based on mineralogy.

| | Coefficients | | | | | Model Summary | | | | | Analysis of Variance | | | | | |
| --- | --- | --- | --- | --- | --- | --- | --- | --- | --- | --- | --- | --- | --- | --- | --- | --- |
| Team | Coef | SE Coef | *t*-Value | *p*-Value | VIF | S | R-sq | R-sq(ad) | R-sq(pred) | Source | DF | Adj SS | Adj MS | *f*-Value | *p*-Value |
| Constant | 980 | 124 | 7.91 | 0.000 | – | 113.511 | 89.11% | 87.71% | 85.48% | Regression | 4 | 3,269,216 | 817,304 | 63.43 | 0.000 |
| log biotite | −446.1 | 643 | −6.94 | 0.000 | 2.25 | – | – | – | – | log biotite | 1 | 620,292 | 620,292 | 48.14 | 0.000 |
| log albite | 286 | 107 | 2.67 | 0.012 | 4.55 | – | – | – | – | log albite | 1 | 92,080 | 92,080 | 7.15 | 0.012 |
| log chlorite | −426.7 | 76.7 | −5.56 | 0.000 | 1.29 | – | – | – | – | log chlorite | 1 | 398,893 | 398,893 | 30.96 | 0.000 |
| log quartz | 332.6 | 84.8 | 3.92 | 0.000 | 4.19 | – | – | – | – | log quartz | 1 | 198,048 | 198,048 | 15.37 | 0.000 |
| – | – | – | – | – | – | – | – | – | – | Error | 31 | 399,428 | 12,885 | – | – |
| – | – | – | – | – | – | – | – | – | – | Total | 35 | 3,668,643 | – | – | – |

The adjusted determination coefficient was 87.71%, and the residue analysis revealed a normal distribution, symmetric histogram, with a very low *p*-value and constant variance for the residues (Figure 10), validating Equation (4).

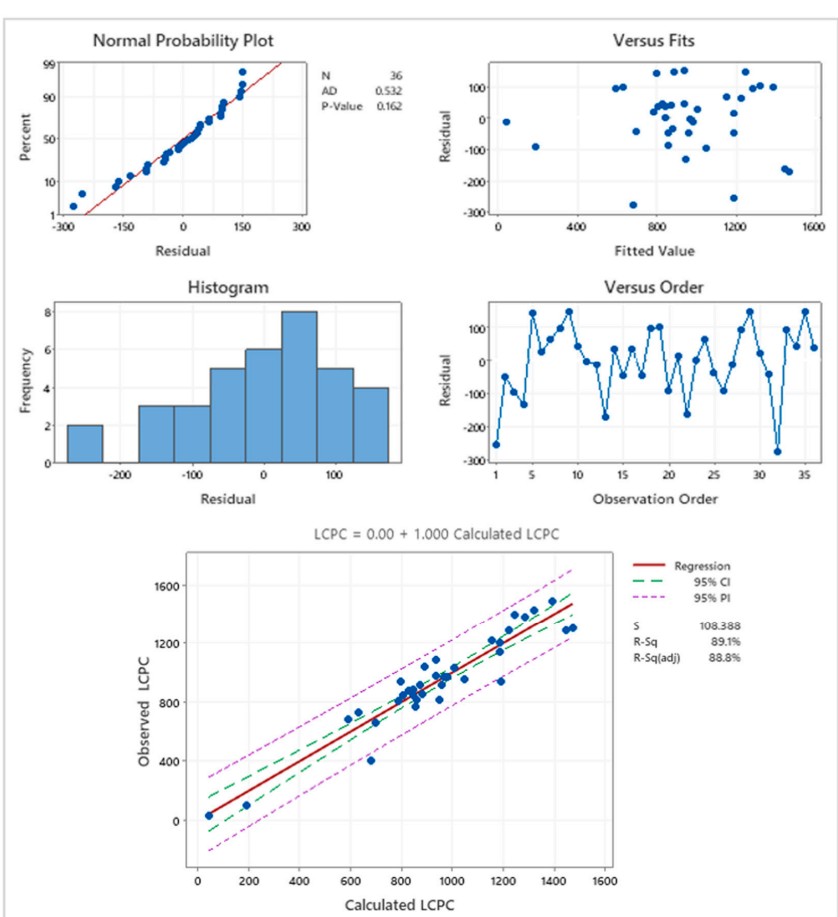

**Figure 10.** Statistical regression from Minitab® [28] to residue analysis on the left top, the correlation between observed versus calculated values of LCPC on the right top and fitted line plot from LCPC data on the bottom.

## 4. Conclusions

The results of the abrasiveness index for the Sossego mine show that abrasiveness varies significantly depending on the lithology, with the MVA being the highest. This information will allow the operation and process team at the Sossego plant to take preventive measures to minimize the impact of processing this highly abrasive material. The validation of the methodology confirmed the similarity of the samples used in the Bond Ai and LCPC tests. The validation also showed that both tests have a low deviation, less than 10%, and the LCPC has an even smaller deviation than Bond's Ai. Finally, this study demonstrated the correlation between Bond Ai and LCPC for the Sossego deposit, thus validating its use for abrasive geometallurgical studies. Such correlation can be validated for other deposits, thus allowing generalized use for geometallurgical studies.

Throughout the work, the observations regarding the variations between the tests pointed to the non-linearity for such aspects: distinct sample support between the tests (mass and grain size interval), different test protocols (rotation speed and residence time), different mechanism of the particle's trajectory inside the respective bowl, and difference in hardness between the standard metal plates of the two tests.

Such results validate the use of the LCPC for geometallurgical studies of the Sossego mine and represent an opportunity for further research, as well as applications aimed at improving the understanding of the LCPC test and its correlation with Bond's Ai for different ores, despite the industry availability of other tests aimed at measuring abrasiveness in rocks.

**Supplementary Materials:** The following are available online at https://www.mdpi.com/article/10.3390/min11121427/s1, Figures S1–S36 include the detailed results of the mineralogical analysis of the samples used for Ai and LCPC tests, including the quantitative mineralogy, the QEMSCAN® images, and a correlation graph of the mineral grade results for each pair of samples.

**Author Contributions:** Conceptualization, P.d.A.B. and M.G.B.; methodology, P.d.A.B.; software, P.d.A.B. and R.K.-R.; validation, P.d.A.B., M.G.B. and E.d.F.; formal analysis, P.d.A.B.; investigation, P.d.A.B.; resources, P.d.A.B. and M.G.B.; data curation, P.d.A.B.; writing—original draft preparation, P.d.A.B., M.G.B., E.d.F. and R.K.-R.; writing—review and editing, P.d.A.B.; visualization, P.d.A.B., M.G.B., E.d.F. and R.K.-R.; supervision, M.G.B.; project administration, P.d.A.B. and M.G.B.; funding acquisition, M.G.B. All authors have read and agreed to the published version of the manuscript.

**Funding:** The abrasiveness tests were carried out through a partnership between VALE-ITV/USP within the Min-AO2 project, Systemic Management of Mine Planning, and Operation at the Mine of the Future, using equipment and labor from the Treatment Laboratory LTM-Poli/USP ore and at Vale's physicochemical laboratories, supported under the Universal CNPQ Project No. 449932/2014-1 and research grant CT2016—308767/2016-0.

**Data Availability Statement:** The geological data and the samples collected, the basis that supported the results, come from the Vale SA Company. Their availability was previously authorized upon consent from the company and through a partnership to finance of the aforementioned project; therefore, the results are not included in its corporate statements in the public domain.

**Acknowledgments:** The authors would like to thank the University of Sao Paulo's Mineral Engineering Post-Graduate Program supported by Maurício Bergerman and the Vale geologist/geometallurgical team, represented by Elisabeth da Fonseca and Rogério Kwitko, for supporting this research project.

**Conflicts of Interest:** The authors declare no conflict of interest.

## Nomenclature

| | |
|---|---|
| IOCG | Iron Oxide Copper and Gold |
| LCPC | Laboratoire Central des Ponts et Chaussées |
| DWT | Drop weight test |
| PLT | Point load test |
| ta | Tumbler index |
| SATMAGAN® | Magnetic susceptibility |

| | |
|---|---|
| Wi | Bond work index |
| Ai | Bond abrasiveness index |
| ACT | Actinolitite |
| BHT | Hydrothermal breccia |
| BIX | Biotite schist |
| BSE | Sequeirinho breccia |
| BSO | Sossego breccia |
| GBA | Gabbro |
| GRA | Granite |
| GRF | Granophyric granite |
| MAG | Magnetitite |
| MVA | Metavolcanic acid |
| TTX | Talc-tremolite schist |
| AMPT | Sample identification |
| HRB | Rockwell Hardness B |
| HB | Brinell Hardness |
| QEMSCAN® | Qualitative evaluation of minerals by scanning electron microscopy |
| SEM | Scanning electron microscopy |
| EDS | Energy-dispersive spectrometry |
| ICP-OES | Inductively coupled plasma—optical emission spectrometry |
| PC | Principal component |
| S | Standard deviation of the distance between data and fitted values |
| R sq | Coefficient of determination |
| R sq(ad) | Adjusted coefficient of determination |
| R sq(pred) | Predicted coefficient of determination |
| DF | Total degrees of freedom |
| SS | Sum of squares |
| MS | Mean squares |
| $f$-Value | Statistic test used to determine whether the term is associated with the response |
| $p$-Value | Probability that measures the evidence against the null hypothesis |
| $t$-Value | Ratio between the coefficient and its standard error |
| SE Coef | Standard error of the coefficient |
| VIF | Variance inflation factor |
| Adj SS | Adjusted sum of squares |
| Adj MS | Adjusted mean squares |

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
