# Peer review of "Determination of Abrasiveness in Copper-Gold Sulfide Ores: A Contribution to the Geometallurgical Model of the Sossego Deposit"

_minerals, doi:10.3390/min11121427_

Round 1

Reviewer 1 Report

General Comments

The paper reports on a small well constrained geometallurgy study for abrasion. The work is well designed and the results look useful. The writing style is clumsy but largely unambiguous. Some specific improvements in grammar and spelling are listed below.

The statistical tables for the regression are extensive and it would be useful to know what statistical package was used to generate these results.

Specific comments

Page 1 line 15 delete “sometimes”

Page 1 line 44 This sentence is deceptive as it implies Peres et al. discussed a range of methods. As far as I can see they only considered the LCPC test.

Page 2 Line 47 “tests” not “testes”

Page 2 line 61 change “which includes the mining open pits” to “which includes the open pits”

Page 3 line 81 “found” not “founded”

“associated to the large’s values ranges given” replace with “and the large range in values provides”

Page 3 line 91 “among” not “along”

Page 3 line 100 “made it necessary” not “made strongly necessary”

Page 3 line 107 “the” not “this”

Page 4 line 128, 129 Replace “as BSE for this study to be the same texture and mineral assemblage” with “to be the same texture and mineral assemblage as BSE in this study”

Page 5 line 159 “were” not “was”

Page 10 line 241 “The data is more adherent when applying a quadratic polynomial” Clumsy grammar. Something more like "The data lies closer to a quadratic curve than a linear trend"

Page 11 line 250 “According” not “Acoordding”

Page 11 Equation 3 Are these parameters really accurate to 5 significant figures?

Author Response

  1. The paper reports on a small well constrained geometallurgy study for abrasion. The work is well designed, and the results look useful. The writing style is clumsy but largely unambiguous. Some specific improvements in grammar and spelling are listed below.

We appreciate your compliments, observations, and support in pointing out the English errors. Before this review, the first text was submitted to an English evaluation by the MDPI. However, some adjustments were made after this review that could lead to new misunderstandings. Now the text has undergone a deep review and a grammatical correction of the final English language. We used your notes to identify the problems and we thank you very much for this.

  1. The statistical tables for the regression are extensive and it would be useful to know what statistical package was used to generate these results.

We used the IoGas for the PCA in Figure 7 and Minitab of regression studies in Figures 9 and 10.

  1. Page 1 line 15 delete “sometimes”

Done.

  1. Page 1 line 44 This sentence is deceptive as it implies Peres et al. discussed a range of methods. As far as I can see they only considered the LCPC test.

Done. We changed the bibliographic reference to Peres' graduation thesis, where she summarized seven methods of abrasiveness measures, among them LCPC and Bond Ai (Table 1).

  1. Page 2 Line 47 “tests” not “testes”

Done

  1. Page 2 line 61 change “which includes the mining open pits” to “which includes the open pits”

Done

  1. Page 3 line 81 “found” not “founded”

Done

  1. “associated to the large’s values ranges given” replace with “and the large range in values provides”

Done

  1. Page 3 line 91 “among” not “along”

Done

  1. Page 3 line 100 “made it necessary” not “made strongly necessary”

Done

  1. Page 3 line 107 “the” not “this”

Done

  1. Page 4 line 128, 129 Replace “as BSE for this study to be the same texture and mineral assemblage” with “to be the same texture and mineral assemblage as BSE in this study”

Done

  1. Page 5 line 159 “were” not “was”

Done

  1. Page 10 line 241 “The data is more adherent when applying a quadratic polynomial” Clumsy grammar. Something more like "The data lies closer to a quadratic curve than a linear trend"

Done

  1. Page 11 line 250 “According” not “Acoordding”

Done

  1. Page 11 Equation 3 Are these parameters really accurate to 5 significant figures?

Done. We adjusted the values in Equation 3 to two decimal places to be significant.

Special thanks!

Reviewer 2 Report

This current investigation is very important for scientific community. For this reason, is necessary to expand the spectrum of the mining situation, and for total validation too. For now, this research is valid only for presented mining situation.

Author Response

  1. This current investigation is very important for scientific community. For this reason, is necessary to expand the spectrum of the mining situation, and for total validation too. For now, this research is valid only for presented mining situation.”

We appreciate your compliments and the emphasis you did on this theme. We agree with the need to observe other environments and we hope to have more approaches in this direction in the future. We're working on it. We have the same tests done on other ores in our own USP laboratory (LTM), which adhere to the correlation seen in this study. We are preparing another paper with these data.

Special thanks!

Reviewer 3 Report

The manuscript appears to be an overview of past tasks as well as adding information that reinforces them. A comparison has been made between LCPC and BOND Ai to determine the practical application of LCPC its advantages and limitations.

It is suggested that the information of other articles be used in the form of a review article with new comparison results with a more open hand.  Therefore, it is suggested that a review article can be written and submitted with more information.

Figures have all been taken from the literature, but it seems permissions have not been requested to editorial offices. While in some cases a scheme can be done from an original one, it is not the case for the data and images. Please check this point.

Technical editing is seriously needed in terms of the scientific English language.

Please find the attachment of the review, including some hand-written points and suggestions. It may be useful.  

Author Response

  1. It is suggested that the information of other articles be used in the form of a review article with new comparison results with a more open hand. Therefore, it is suggested that a review article can be written and submitted with more information.

We appreciate your comments. Our intention was, in fact, to make a correlation study and evaluate abrasiveness between the different types of rocks and mineralogical compositions in the same mineral deposit and that it contained such variability. It really wasn't our intention to prepare a review article aggregating other articles based on this same validation. However, our USP research group already has a database of this comparison with other ores that we plan to publish and we also plan to work on a review paper.

  1. Figures have all been taken from the literature, but it seems permissions have not been requested to editorial offices. While in some cases a scheme can be done from an original one, it is not the case for the data and images. Please check this point.

We had explained this misconception of the references in Figures 1 and 2 to Nikolas Burazer as you can see in the email below:

To clarify, we removed the citation from these two Figures. This work was supported by Vale, whose terms of the project involved have been described in the "Funding" comments.

  1. Technical editing is seriously needed in terms of the scientific English language.

We appreciate your compliments, observations, and support in pointing out the English errors. Before this review, the first text was submitted to an English evaluation by the MDPI. However, some adjustments were made after this review that could lead to new misunderstandings. Now the text has undergone a deep review and a grammatical correction of the final English language. We used your notes to identify some of these problems and we thank you very much for this.

  1. Please find the attachment of the review, including some hand-written points and suggestions. It may be useful.

About your marks, we:

  • We removed reference 22 in the Figures 1 and 2;
  • We input reference 22 in the text;
  • We input the new reference 25 from Google Earth, with the reference image date;
  • We evaluated your suggestion to normalize the LCPC, but it is worth mentioning that the Bond Ai data are not normalized. We carried out simulations with normalized data, but we didn't see any difference even in the equations, only that some data began to turn negative and this made visual representation in the correlation charts difficult. That being the case, we kept the primary data and emphasized that the two groups of data are raw and non-normalized;
  • We changed the graph names in Figure 6;
  • We improved Figure 9b by resizing and modifying the layout.
  • We improved Figures 10a and 10b.

Reviewer 4 Report

This is one of the best papers I've read in terms of geomet and the assistance of mineralogy in such a long time.  The English is fine.  I'd request that the Axis on graphs are fully labelled please.    

Details of the tests, samples, masses, objectives were great.  Details of the Qemscan was excellent.  The only thing I would suggest to insert is the details such as accelerating voltage, spot size etc.  It would complete a comprehensive description.  I'd love to see some description on sample preparation as your QEMscan images clearly show 'fine' to coarse particles.  Did you have issues with density distribution during prep?  Maybe writing in a few comments on this area would be useful.

Figure 9's and other ioGAS(?) plots could use better resolution to read all the labels.

Unless I missed it, are there any comments on using 135 samples alone?  What is the representivity across the pit, other than just concluding that there is slight variations within each lithology?

More work should be done like this on establishing the 'added value' of mineralogy and benchtop tests to the likes of prevention, prediction and less of the forensic use.

Author Response

  1. This is one of the best papers I've read in terms of geomet and the assistance of mineralogy in such a long time. The English is fine. I'd request that the Axis on graphs are fully labelled please.

We appreciate your comments and I hope I have met your expectations on this geomet study. We worked on the graphs to fix these problems.

  1. The only thing I would suggest to insert is the details such as accelerating voltage, spot size etc. It would complete a comprehensive description. I'd love to see some description on sample preparation as your QEMscan images clearly show 'fine' to coarse particles. Did you have issues with density distribution during prep? Maybe writing in a few comments on this area would be useful.

We managed to insert more information about preparation to meet your need. You will see some of the details in the referenced articles because we used the same methodology as described there.

  1. Figure 9's and other ioGAS(?) plots could use better resolution to read all the labels.

We improved the quality of Figure 9 and, yes, it was made by ioGAS.

  1. Unless I missed it, are there any comments on using 135 samples alone? What is the representivity across the pit, other than just concluding that there is slight variations within each lithology?

The 135 samples were used for the geometallurgical study that was done during the project development and covered the mining plans from 2004 until 2017. No comparative study including the historical data could be carried out at the authors’ option, given the differences in the type of samples used and doubts about the lithological classification protocols and sample handling and preparation for the Bond Ai of historical geometallurgical campaigns. We did some changes on lines 62 – 73 and 113-117 in order to clarify this point. Please confirm if you think any additional adjustment in the paper is necessary.

  1. More work should be done like this on establishing the 'added value' of mineralogy and benchtop tests to the likes of prevention, prediction and less of the forensic use.

We appreciate your comments and will work on this in future studies to keep the work development in this line of research.

Special thanks!

Round 2

Reviewer 3 Report

I hope to see your publication soon. As a minor revise, please consider the below points.

I suggest that you provide images in linear and better quality to improve the article for publishing.

In the supplementary info section, a brief description of the images for quick understanding of the readers; nomenclature for symptoms and abbreviations can also help to understand your work better and faster.

In the main text of the article, try to have a Nomenclature section similar to the article below to give a better understanding of the signs and abbreviations.

Bioleaching of copper from chalcopyrite ore at higher NaCl concentrations - ScienceDirect

https://www.sciencedirect.com/science/article/abs/pii/S0892687521005100

Author Response

Dear Dr. Reviewer,

First of all, we would like to thank the editors and reviewers of this study who presented valuable suggestions and pertinent comments, thus improving the quality of our article. After a detailed analysis of the comments and questions, as well as responding to the errors pointed out and suggestions contained in the opinions sent to us, the article underwent changes, which are indicated below.

I suggest that you provide images in linear and better quality to improve the article for publishing.

We adjusted the sharpness of figures 1 to 10 and changed all figures from the appendix A (figures A1 to A36). We understand that now they are better for the publication.

In the supplementary info section, a brief description of the images for quick understanding of the readers; nomenclature for symptoms and abbreviations can also help to understand your work better and faster.

The following paragraph was included: “The data presented in the Appendix A, figures A1 to A36, includes the detailed results of the mineralogical analysis of the samples used for Ai and LCPC tests, including, the quantitative mineralogy, the Qemscan images and a correlation graph of the mineral grade results for each pair of samples.”

The list of nomenclatures was included in the main text.

In the main text of the article, try to have a Nomenclature section similar to the article below to give a better understanding of the signs and abbreviations.

The nomenclature section was included in page 2.

See attached the revised paper with all the corrections.

Special thanks!
